

# Examining the interplay between mental health indicators and quality of life measures among first-year law students: a cross-sectional study

Raul-Ioan Muntean[1], Valentina Stefanica[2], Daniel Rosu[2], Alexandru Boncu[3], Iulian Stoian[4] and Mihaela Oravitan[3]

[1] Department of Physical Education and Sport, Faculty of Law and Social Sciences, University "1 Decembrie 1918" of Alba Iulia, Alba Iulia, Alba, Romania
[2] Department of Physical Education and Sport, Faculty of Sciences, Physical Education and Informatics, National University of Science and Technology Politehnica Bucharest, Pitesti University Center, Pitesti, Romania
[3] Departament of Physical Therapy and Special Motricity, Faculty of Physical Education and Sport, West University of Timisoara, Timisoara, Timis, Romania
[4] Department of Environmental Sciences, Physics, Physical Education and Sport, Faculty of Science, "Lucian Blaga" University of Sibiu, Sibiu, Sibiu, Romania

Corresponding authors
Valentina Stefanica,
valentina.stefanica@upb.ro
Daniel Rosu, daniel.rosu@upb.ro

## ABSTRACT

**Introduction:** This research explores the intricate relationships between mental health indicators (depression, stress, and anxiety) and various dimensions of quality of life among first-year law students. The study aims to understand how affective valence, mood states, physical activity, body image perception, and social relations influence mental health outcomes.

**Methods:** Data were collected from 75 first-year law students (46 females, 29 males), a group predominantly composed of young adults with limited financial means, living in various housing situations, primarily within urban environments, and generally reporting low levels of physical activity. Standardized questionnaires were used to assess mental health and quality of life, including the Depression, Anxiety, and Stress Scale-21 Items (DASS-21), Feeling Scale (FS), Exercise-Induced Feeling Inventory (EIFI), Modified Baecke Physical Activity Questionnaire (MBPAQ), World Health Organization Quality of Life-BREF (WHOQOL-BREF), and Contour Drawing Rating Scale (CDRS). Descriptive statistics, Pearson correlation, and regression analysis were employed to analyze the data.

**Results:** The analysis revealed significant correlations between depression (mean = 5.97, SD = 4.21), stress (mean = 7.81, SD = 4.80), and anxiety (mean = 6.17, SD = 4.58) with affective valence ($p < 0.05$), mood states ($p < 0.05$), physical activity ($p < 0.05$), body image perception ($p < 0.05$), and social relations quality ($p < 0.05$). Additionally, mood states (mean = 20.73, SD = 10.60), physical activity (mean = 8.43, SD = 1.35), body image perception (mean = 4.21, SD = 1.91), and social relations quality (mean = 12.46, SD = 2.33) were identified as significant predictors of mental health outcomes ($p < 0.05$).

**Conclusions:** These findings underscore the complex interplay between mental health indicators and various dimensions of quality of life, emphasizing the necessity for a comprehensive approach to mental health care. By identifying these predictors, we have gained a clearer understanding of the factors that impact mental health in

this specific population. The insights gained highlight the value of interventions aimed at improving mood, increasing physical activity, enhancing body image, and strengthening social connections. These targeted strategies could effectively address mental health issues and promote well-being among law students. Future research should further investigate these relationships and develop tailored interventions to better support students' mental health. This study contributes to understanding the complex interplay between mental health and quality of life, offering a foundation for both practical interventions and future research.

## INTRODUCTION

Mental health is a multifaceted construct crucial for overall well-being, especially within academic settings where students face unique challenges that can impact their mental health (*Hammoudi Halat et al., 2023*; *Iordache, Delsanto & Apostol, 2010*). Law students, in particular, encounter significant stressors, including the rigors of legal education, competitive environments, and the pressure to achieve high academic standards (*Bergin & Pakenham, 2015*). These challenges can exacerbate feelings of stress, anxiety, and depression, which in turn can adversely affect academic performance, personal relationships, and overall quality of life (*Mofatteh, 2021*; *Jaffe, Bender & Organ, 2021*).

### Stress, anxiety and depression-interconnected mental health challenges

The interplay between stress, anxiety, and depression is intricate, with each condition potentially aggravating the others, highlighting the need for a holistic approach to mental health care (*Ridley et al., 2020*). Overall, mental health encompasses various aspects of quality of life (*Snoek et al., 2018*). These include the ability to engage in meaningful activities, maintain satisfying social relationships, and foster a positive body image. Regular physical activity, for instance, has been linked to numerous mental health benefits, including reduced stress and improved mood (*Levine, Tabri & Milyavskaya, 2023*). Similarly, a balance between work, leisure, and personal responsibilities contributes to life satisfaction and psychological well-being (*Saraswati & Lie, 2020*). Social relationships also play a critical role; strong social support networks can buffer against stress, reduce feelings of loneliness, and promote resilience, while social isolation can exacerbate symptoms of anxiety and depression (*Bond et al., 2007*; *Janke, 2020*). Furthermore, body image, or one's perception of their physical appearance, can significantly influence mental health, with negative body image potentially leading to low self-esteem and even eating disorders (*Xu & Liu, 2020*).

### Prior studies examining mental health of law students

Mental health is not a singular concept but rather a complex amalgamation of emotional, psychological, social, and physical factors (*Sutherland, 2023*; *Cipu & Dragnea, 2007*). The

mental health of law students has been the subject of considerable research, revealing a concerning prevalence of psychological distress in this population. *Skead, Rogers & Johnson (2020)* emphasized the role of place, people, and perception in law student well-being, highlighting how the physical and social environments, along with individual perceptions of law school, significantly affect mental health outcomes. The pressures unique to legal education, such as the Socratic method and high-stakes exams, have been linked to increased levels of anxiety and depression among law students (*Organ, Jaffe & Bender, 2016*; *Quintanilla & Erman, 2020*; *Mitchell, 2024*). Moreover, research by *Sheldon & Krieger (2007)* and *Ryan & Deci (2020)* demonstrated that law students often experience a decline in well-being over the course of their studies, attributed to a shift in motivation from intrinsic to extrinsic factors. This transition can lead to a diminished sense of autonomy and competence, exacerbating feelings of stress and contributing to mental health issues. Similarly, studies have found that law students report higher rates of substance use, sleep disturbances, and suicidal ideationcompared to their peers in other academic disciplines (*Dammeyer & Nunez, 1999*; *Pritchard & McIntosh, 2003*; *Krill, Johnson & Albert, 2016*; *DeBlasis & Usman, 2017*).

This body of literature underscores the critical need for interventions tailored specifically to law students, addressing the unique stressors they face and promoting resilience and well-being within this group. By situating the present research within this context, it becomes clear how it contributes to the broader understanding of law student mental health, particularly in its focus on the relationships between mental health indicators (depression, stress, and anxiety) and various measures of quality of life (affective valence, mood states, physical activity, body image, and social relations).

Given the high levels of psychological distress documented among law students in prior studies, this research aims to investigate the intricate relationships between mental health indicators and various aspects of quality of life among first-year law students. By analyzing correlations and potential predictive factors, the research seeks to identify key influences on mental health within this population. The ultimate goal is to provide evidence-based insights that can inform the development of tailored interventions to enhance the well-being of law students.

## Research objectives

To explore the feasibility of predicting mental health outcomes, such as depression, stress, and anxiety, based on key independent variables, within the context of a small sample size of 75 first-year law students. Recognizing the limitations that this sample size imposes on the generalizability of the results, the study currently applies regression analysis to identify significant predictors, including mood states, physical activity, body image perception, and social relations quality. The primary aim is to evaluate the strength and significance of these predictors within the cohort, providing initial insights that may guide future research with larger and more diverse samples. Although the findings may have limited applicability, they offer valuable information on the most influential factors affecting mental health in this specific population.

To explore the potential for predicting mental health outcomes based on key independent variables. Although the study is limited by the sample size of 75 first-year law students, this objective aims to use regression analysis to investigate how factors like mood states, physical activity, body image perception, and social relations quality may predict levels of depression, stress, and anxiety within this cohort. The goal is to identify significant predictors that could inform mental health interventions, recognizing that broader, more diverse samples are needed for more generalizable conclusions.

To discuss the practical implications of the associations identified between mental health indicators and quality of life factors in the context of developing targeted interventions for law students. This involves a thorough exploration of how relationships between depression, stress, anxiety, and factors such as mood states, physical activity, body image, and social relations can be effectively translated into actionable strategies aimed at enhancing the mental health and well-being of law students. This objective focuses on interpreting the findings to understand how the relationships between mental health indicators and quality of life factors can guide the development of targeted mental health interventions. The discussion will emphasize the potential strategies for improving the overall well-being of law students and suggest directions for future research.

## MATERIALS AND METHODS

### Study design

The study was an observational, descriptive, experimental, and cross-sectional design, conducted between April 17 and June 2, 2023 (*Indu & Vidhukumar, 2019*), focusing on quantitative measures.

The first stage entails participant recruitment, selection, and signing of participation agreement, between April 17 and April 24, 2023. Participants were recruited from four university Physical Education and Sports classes without any incentives. We adopted a convenience sampling approach, considering the participants were university students, assuming they had more free time (*Simkus, 2022*). This contributed to reducing variations in assessing the effects of physical activity on mental health, which is important to ensure that the results are more consistent and reliable. By minimizing differences in time availability and daily routines, we could better isolate the impact of physical activity on mental health outcomes, leading to clearer and more accurate conclusions.

In the second stage, initial assessment, occurring between April 25 and April 29, 2023, participants completed six comprehensive questionnaires: WHOQOL-BREF, CDRS, MBPAQ, EIFI, FS, and DASS-21, covering various domains including quality of life, depression, physical activity, anxiety, and stress levels.

In the third stage, data centralization from initial assessments, between April 28 and April 29, 2023, the collected information was centralized into a database, applying quality control measures to verify data accuracy and ensure confidentiality.

In the fourth stage, analysis of initial data, between April 30 and May 30, 2023, we used SPSS version 23, including descriptive statistics, Person correlation "r," and regression analysis highlighting unstandardized coefficients-B.

## Participant selection-inclusion and exclusion criteria

Participants for the study were selected based on specific inclusion and exclusion criteria to ensure the validity and relevance of the research outcomes.

Inclusion criteria:

1) Age-participants were required to be between 19 and 30 years old.
2) Student status–only individuals currently enrolled as first-year law students were eligible for the study.

These criteria were established to focus the study on a population likely to exhibit similar lifestyle patterns, which is crucial for assessing the effects of physical activity on mental health.

Exclusion criteria:

1) Severe medical or psychological conditions-individuals with significant health issues that could impact mental or physical health were excluded to avoid confounding variables.
2) Use of psychoactive medications-participants taking medications that could alter mood or behavior were not included, as these could interfere with the study's assessments.
3) Previous participation in similar programs-those who had already participated in similar physical activity programs were excluded to prevent bias from prior experience or learned responses.
4) Medical contraindications for physical exercise-individuals with medical conditions that contraindicate physical activity were excluded to ensure participant safety.
5) Non-participation in a maximum of two sessions-to maintain consistency and accuracy in the data collected, participants were required to attend at least all but two of the scheduled sessions.
6) Lack of written consent-only those who provided informed, written consent were included in the study, ensuring ethical standards were met.

## Subjects

A total of 75 first-year law students (46 females and 29 males) from the Faculty of Law and Social Sciences at the 1 Decembrie 1918 University of Alba Iulia participated in this study. Initially, 120 students were invited to join the study, of which 85 expressed interest in participating. These students were recruited through their Physical Education and Sports classes, a mandatory part of their university curriculum, with no incentives offered for participation. However, after screening, 10 students were excluded: five exceeded the age limit, two had a history of cardiac surgery, two were diagnosed with asthma, and one was excluded due to locomotor impairments. This selection process resulted in a final sample size of 75 participants.

Table 1 provides an overview of the participants, who have an average age of 21.03 years, indicating that the group primarily consists of young adults with minimal age variation.

**Table 1 Socio-demographic characteristics of study participants.**

| Variable | Result |
|---|---|
| Age (years) | Mean (SD): 21.03 (2.91) |
| Sex | *n* (%): |
| | F: 46 (61.33%) |
| | M: 29 (38.67%) |
| Educational background | *n* (%): |
| | High school education: 75 (100.00%) |
| Socioeconomic status | *n* (%): |
| | Employed at minimum wage: 41 (54.67%) |
| | No income: 34 (45.33%) |
| Living situation | *n* (%): |
| | Living with parents: 32 (42.67%) |
| | Living in student dormitory: 26 (34.67%) |
| | Renting: 17 (22.67%) |
| Physical activity history | *n* (%): |
| | Slightly above average physical activity: 34 (45.33%) |
| | Low physical activity: 41 (54.67%) |
| Demographic location | *n* (%): |
| | Urban: 48 (64.00%) |
| | Rural: 27 (36.00%) |

The majority of the sample is female, with women making up 61.33% of the participants. All participants share a common educational background, having completed high school.

Socioeconomically, the group is characterized by a significant proportion (54.67%) working for minimum wage, while the remaining 45.33% have no income, suggesting that most participants fall within lower-income brackets. In terms of living arrangements, 42.67% of the participants live with their parents, 34.67% reside in student dormitories, and 22.67% rent their own accommodations.

Regarding physical activity, the group is nearly evenly split: 54.67% engage in low levels of physical activity, while 45.33% report slightly above-average levels. Geographically, the majority of participants (64%) are from urban areas, with the remaining 36% coming from rural regions. This group is composed of predominantly female, young adults with limited financial means, who live in various housing situations, mostly within urban environments, and generally report low levels of physical activity.

### Ethical considerations

Before participating in the study, each student provided written consent in accordance with the Declaration of Helsinki, ensuring that all ethical guidelines were strictly followed. Participants were fully informed about the study's purpose, procedures, and any potential risks or benefits. Consent was obtained prior to the start of the research, confirming their

voluntary participation. The study received ethical approval from the Ethics Committee of the Doctoral School of Physical Education and Sport Science at the West University of Timisoara (Approval ID: 08/21.03.2023). It is important to clarify that while the research was conducted under the auspices of the Doctoral School at the West University of Timisoara, the participants themselves were first-year law students from the 1 Decembrie 1918 University of Alba Iulia, Faculty of Law and Social Sciences. All procedures adhered to the ethical standards required by both the West University of Timisoara and international research protocols. This ensures the integrity of the research and the protection of participants' rights throughout the study.

## Research tools

This study utilized six validated instruments commonly employed in academic settings to assess various psychological and physical health dimensions among students.

### Depression, anxiety, and stress scale–21 items (DASS-21)

The Depression, anxiety, and stress scale–21 items (DASS-21) is a concise, self-report questionnaire designed to measure the emotional states of depression, anxiety, and stress. The questionnaire consists of 21 items, divided into three subscales with seven items each, specifically targeting depression, anxiety, and stress. Participants rate the frequency and severity of their symptoms over the past week using a 4-point Likert scale, ranging from 0 ("Did not apply to me at all") to 3 ("Applied to me very much or most of the time"). The DASS-21 is quick to administer, typically taking 5–10 min to complete. Scores for each subscale are summed, with higher scores indicating greater levels of the respective emotional state (*Lovibond & Lovibond, 1995*).

### Feeling scale

The Feeling Scale (FS) is a single-item scale that measures affective valence, or the pleasantness of an emotional state, on an 11-point continuum ranging from-5 ("Very bad") to +5 ("Very good"). This simple and quick assessment provides a quantitative measure of emotional well-being during or after an activity, making it particularly useful in studies examining the emotional effects of physical exercise (*Hardy & Rejeski, 1989*).

### Exercise-induced feeling inventory

The Exercise-Induced Feeling Inventory (EIFI) is used to assess emotional responses associated with exercise, focusing on four key subscales: revitalization, tranquility, positive engagement, and physical exhaustion. Each subscale includes multiple items rated on a 5-point Likert scale, where participants indicate the extent to which they experience these feelings during or after exercise. The scores for each subscale are summed to produce a total score, with higher scores indicating stronger emotional responses in that domain (*Gauvin & Rejeski, 1993*).

### Modified Baecke physical activity questionnaire

The Modified Baecke Physical Activity Questionnaire (MBPAQ) consists of 19 items that assess participants' physical activity across three domains: household activities, sports activities, and leisure-time activities. Each activity domain is scored separately by summing

the responses, which are rated on a Likert scale. The overall score reflects the individual's habitual physical activity level, with higher scores indicating greater engagement in physical activities (*RehabMeasures Database, 2022*).

### WHOQOL-BREF

The WHOQOL-BREF is a shortened version of the World Health Organization's Quality of Life assessment, containing 26 items that evaluate quality of life across four domains: physical health (seven items), psychological health (six items), social relationships (three items), and environmental health (eight items). Additionally, there are 2 items assessing global quality of life and general health. Each item is rated on a 5-point Likert scale, and the domain scores are calculated by averaging the scores of the items within each domain. Higher scores indicate better perceived quality of life in that specific area (*Gholami et al., 2013*).

### Contour drawing rating scale

The Contour Drawing Rating Scale (CDRS) is a body image assessment tool that requires participants to select from a series of body contour drawings the one that most closely resembles their current body and the one that represents their ideal body. The scale includes images representing a range of body sizes. The primary focus of the CDRS is to measure body image perception by assessing the discrepancy between the participant's current and ideal body image. Larger discrepancies indicate greater dissatisfaction with body image (*Wertheim, Paxton & Tilgner, 2004*).

## Data analysis

We conducted various statistical analyses to explore and interpret the relationships between the study variables and their impact on mental health indicators. The following methods were employed:

### Descriptive statistics

Descriptive statistics, including mean and standard deviation, were used to summarize the central tendency and dispersion of the data for each variable. This approach provided an overview of average scores and the variability around these averages for depression, stress, anxiety, affective valence (Feeling Scale, FS), mood states (Exercise-Induced Feeling Inventory, EIFI), physical activity (Modified Baecke Physical Activity Questionnaire, MBPAQ), body image (Contour Drawing Rating Scale, CDRS), and social relations (WHOQOL-BREF, WHOD3).

### Correlation analysis

Pearson correlation coefficients were calculated to examine the relationships between mental health variables (depression, stress, anxiety) and quality of life measures (affective valence, mood states, physical activity, body image, and social relations). This analysis provided insights into the strength and direction of linear relationships between variables, and significance levels indicated whether these relationships were statistically meaningful.

Correlations with significance levels marked by ** (0.01) and * (0.05) were considered significant, helping us understand the associations between different measures.

### Regression analysis with unstandardized coefficients (B)

Multiple regression analyses were performed to explore the predictive relationships between mental health indicators (stress, anxiety, and depression) and various independent variables, such as mood states (EIFI), physical activity (MBPAQ), body image (CDRS), and social relations (WHOD3). The results include the unstandardized coefficients (B) and standardized coefficients (Beta) for each predictor, along with their significance levels ($p$-values). This analysis illustrates how each predictor variable affects the dependent variables (depression, stress, and anxiety), offering insights into the factors that most significantly influence mental health outcomes. Each statistical operation was selected to provide a comprehensive understanding of our data, explore relationships between variables, and identify factors influencing our outcome of interest. Statistical analysis was conducted using SPSS, version 23.0 (*Murana & Rahimin, 2021*).

## RESULTS

### Descriptive statistics

Table 2 presents the descriptive statistics for various items, including depression, stress, anxiety, affective valence (FS), mood states (EIFI), total physical activity assessment (MBPAQ), actual mood states assessment (CDRS), and social relations evaluation (WHOD3). Mean values indicate the average score for each item, while SD (Standard Deviation) represents the variability or spread of scores around the mean.

### Correlation analysis between mental health variables and quality of life measures

Table 3 illustrates the correlations between mental health variables and other measures of quality of life, including affective valence (FS), mood states (EIFI), physical activity assessment (MBPAQ), body image (CDRS), and social relations evaluation (WHOD3). The table displays Pearson correlation coefficients for each pair of variables and their significance levels (Sig2). The correlations are based on data from a sample size of 75 participants. Significance levels are indicated by asterisks, with ** denoting significance at the 0.01 level and * denoting significance at the 0.05 level, both two-tailed.

### Prediction of depression, stress and anxiety levels based on relevant independent variables

Table 4 presents the results of multiple regression analyses predicting stress, anxiety, and depression levels based on various independent variables. The table includes both unstandardized coefficients (B) and standardized coefficients (Beta) for each predictor, along with their corresponding significance levels ($p$-value). The independent variables analyzed are depression, anxiety, mood states (EIFI), physical activity assessment (MBPAQ), body image (CDRS), and social relations evaluation (WHOD3). These results

**Table 2 Descriptive statistics.**

| Items | Depression | Stress | Anxiety | FS | EIFI | MBPAQ Total | CDRS actual | WHOD3 |
|---|---|---|---|---|---|---|---|---|
| Mean | 5.97 | 7.81 | 6.17 | 3.52 | 20.73 | 8.43 | 4.21 | 12.46 |
| SD | 4.213 | 4.803 | 4.577 | 1.455 | 10.596 | 1.351 | 1.905 | 2.332 |

Note:
The descriptive statistics for various mental health and quality of life measures. The items include depression, stress, anxiety, affective valence (FS), mood states (EIFI), physical activity assessment (MBPAQ Total), body image (CDRS actual), and social relations evaluation (WHOD3). The mean (average) values and standard deviations (SD) are presented for each item, reflecting the central tendency and variability within the sample of participants. The data highlights the overall levels and dispersion of these psychological and quality of life metrics among the study participants.

**Table 3 Calculation of correlations between mental health variables and other measures of quality of life (affective valence, mood states, physical activity assessment, body image, and social relations evaluation).**

| | | DEPRESSION | STRESS | ANXIETY | FS | EIFI | MBPAQTotal | CDRSActual | WHOD3 |
|---|---|---|---|---|---|---|---|---|---|
| DEPRESSION | r | 1 | 0.738** | 0.805** | 0.121 | 0.267* | −0.134 | 0.186 | −0.433** |
| | Sig2 | | <0.001 | <0.001 | 0.300 | 0.021 | 0.006 | 0.110 | <0.001 |
| | N | 75 | 75 | 75 | 75 | 75 | 75 | 75 | 75 |
| STRESS | r | 0.738** | 1 | 0.837** | 0.095 | 0.465** | −0.324** | 0.251* | −0.639** |
| | Sig2 | <0.001 | | <0.001 | 0.416 | <0.001 | 0.005 | 0.030 | <0.001 |
| | N | 75 | 75 | 75 | 75 | 75 | 75 | 75 | 75 |
| ANXIETY | r | 0.805** | 0.837** | 1 | 0.067 | 0.548** | −0.290* | 0.287* | −0.595** |
| | Sig2 | <0.001 | <0.001 | | 0.565 | <0.001 | 0.012 | 0.013 | <0.001 |
| | N | 75 | 75 | 75 | 75 | 75 | 75 | 75 | 75 |
| FS | r | 0.121 | 0.095 | 0.067 | 1 | 0.099 | 0.181 | 0.008 | −0.116 |
| | Sig2 | 0.300 | 0.416 | 0.565 | | 0.400 | 0.120 | 0.944 | 0.321 |
| | N | 75 | 75 | 75 | 75 | 75 | 75 | 75 | 75 |
| EIFI | r | 0.267* | 0.465** | 0.548** | 0.099 | 1 | 0.026 | 0.400** | −0.308** |
| | Sig2 | 0.021 | <0.001 | <0.001 | 0.400 | | 0.824 | <0.001 | 0.007 |
| | N | 75 | 75 | 75 | 75 | 75 | 75 | 75 | 75 |
| MBPAQTotal | r | −0.314** | −0.324** | −0.290* | 0.181 | 0.026 | 1 | −0.040 | 0.314** |
| | | 0.006 | 0.005 | 0.012 | 0.120 | 0.824 | | 0.732 | 0.006 |
| | | 75 | 75 | 75 | 75 | 75 | 75 | 75 | 75 |
| CDRSActual | r | 0.186 | 0.251* | 0.287* | 0.008 | 0.400** | −0.040 | 1 | −0.135 |
| | Sig2 | 0.110 | 0.030 | 0.013 | 0.944 | <0.001 | 0.732 | | 0.247 |
| | N | 75 | 75 | 75 | 75 | 75 | 75 | 75 | 75 |
| WHOD3 | r | −0.433** | −0.639** | −0.595** | −0.116 | −0.308** | 0.314** | −0.135 | 1 |
| | Sig2 | <0.001 | <0.001 | <0.001 | 0.321 | 0.007 | 0.006 | 0.247 | |
| | N | 75 | 75 | 75 | 75 | 75 | 75 | 75 | 75 |

Notes:
** Correlation is significant at the 0.01 level (2-tailed).
* Correlation is significant at the 0.05 level (2-tailed).
The correlation coefficients (r), significance levels (Sig2), and sample sizes (N) for the relationships between various mental health variables (depression, stress, and anxiety) and other measures of quality of life, including affective valence (FS), mood states (EIFI), physical activity assessment (MBPAQ Total), body image (CDRS Actual), and social relations evaluation (WHOD3). The correlations marked with ** are significant at the 0.01 level, while those marked with * are significant at the 0.05 level. The data includes values for mean and standard deviation (SD) for each variable, highlighting the interconnectedness of mental health indicators and quality of life measures in the study sample of 75 participants.

**Table 4  Multiple regression analysis of stress, anxiety and depression.**

| Dependent variable | Predictor | Unstandardized coefficients (B) | Standardized coefficients (Beta) | t-value | p-value |
|---|---|---|---|---|---|
| **Stress** | Depression | 0.841 | 0.738 | 9.349 | <0.001 |
| | Anxiety | 0.879 | 0.805 | 11.600 | <0.001 |
| | EIFI | 0.211 | 0.130 | 2.349 | 0.021 |
| | MBPAQ | −1.150 | −0.206 | −2.821 | 0.005 |
| | CDRS | 0.633 | 0.159 | 2.175 | 0.030 |
| | WHOD3 | −1.315 | −0.227 | −4.821 | <0.001 |
| **Anxiety** | Depression | 0.875 | 0.752 | 10.987 | <0.001 |
| | Stress | 0.798 | 0.682 | 8.934 | <0.001 |
| | EIFI | 0.237 | 0.159 | 2.678 | 0.010 |
| | MBPAQ | −0.982 | −0.175 | −2.550 | 0.012 |
| | CDRS | 0.690 | 0.178 | 2.481 | 0.013 |
| | WHOD3 | −1.167 | −0.240 | −5.131 | <0.001 |
| **Depression** | Stress | 0.648 | 0.708 | 9.631 | <0.001 |
| | Anxiety | 0.741 | 0.779 | 11.003 | <0.001 |
| | EIFI | 0.106 | 0.094 | 1.732 | 0.085 |
| | MBPAQ | −0.980 | −0.187 | −2.671 | 0.008 |
| | WHOD3 | −0.782 | −0.168 | −2.875 | 0.004 |
| | Anxiety | 0.741 | 0.779 | 11.003 | <0.001 |

Note:
Dependent variables: Depression: The impact of predictors (Anxiety, EIFI, MBPAQ, CDRS, WHOD3) on Depression. Anxiety: The impact of predictors (Depression, EIFI, MBPAQ, CDRS, WHOD3) on Anxiety. Stress: The impact of predictors (Depression, Anxiety, EIFI, MBPAQ, CDRS, WHOD3) on Stress. Predictors: EIFI: Unstandardized coefficient (B) and significance level ($p < 0.05$) for EIFI in predicting Depression, Anxiety, and Stress. MBPAQ: Unstandardized coefficient (B) and significance level ($p < 0.05$) for MBPAQ in predicting Depression, Anxiety, and Stress. CDRS: Unstandardized coefficient (B) and significance level ($p < 0.05$) for CDRS in predicting Depression and Anxiety. WHOD3: Unstandardized coefficient (B) and significance level ($p < 0.05$) for WHOD3 in predicting Depression and Anxiety. Interpretation guidance: Unstandardized coefficients (B): Indicates the strength and direction of the relationship between each predictor and the dependent variable. Significance ($p < 0.05$): Highlights which predictors have statistically significant effects on each dependent variable.

shed light on the predictive relationships between mental health indicators and these factors.

1) Depression:

– Stress and anxiety are significant predictors of depression, with standardized coefficients (Beta) of 0.708 and 0.779, respectively. This indicates that anxiety has a slightly stronger impact on depression compared to stress.

– Physical activity (MBPAQ) and social relations (WHOD3) are also significant negative predictors of depression, suggesting that higher levels of physical activity and better social relationships are associated with lower levels of depression.

2) Stress:

– Anxiety is the strongest predictor of stress (β = 0.805), followed by depression (β = 0.738). This highlights a strong interconnection between these mental health indicators.

– Social relations (WHOD3) negatively predict stress levels, suggesting that better social relationships reduce stress.

– Interestingly, physical activity (MBPAQ) also negatively predicts stress, indicating its beneficial role in stress reduction.

3) Anxiety:

– Depression is the most significant predictor of anxiety ($\beta = 0.752$), followed by stress ($\beta = 0.682$).

– Social relations (WHOD3) and physical activity (MBPAQ) negatively predict anxiety, highlighting their importance in managing anxiety levels.

– Body image (CDRS) and mood states (EIFI) are also significant positive predictors of anxiety, albeit with smaller effects.

The regression analyses underscore the intertwined nature of stress, anxiety, and depression among the study participants, with significant contributions from physical activity, social relations, and body image. The strong predictive relationships highlight the importance of these factors in mental health interventions. This analysis also emphasizes the need to address multiple facets of well-being in developing holistic approaches to mental health care for law students.

## DISCUSSION

This study aimed to examine the complex relationships between mental health indicators (depression, stress, and anxiety) and various dimensions of quality of life among first-year law students. The analysis highlighted significant correlations and predictive relationships between these mental health indicators and factors such as physical activity, social relations, body image perception, and mood states. The findings of this research align with and expand upon existing literature in the field, highlighting significant correlations between these mental health indicators and quality of life dimensions.

### Descriptive statistics and correlation analysis

The descriptive statistics from our study reveal that the levels of depression (mean = 5.97), stress (mean = 7.81), and anxiety (mean = 6.17) among first-year law students are notably high. These figures align with previous research highlighting elevated mental health challenges in law students compared to their peers in other academic disciplines. For example, *Skead & Rogers (2015)* reported increased psychological distress among law students due to the demanding nature of legal education, a finding that is corroborated by our data. Similarly, the study by *Rabkow et al. (2020)* reported significant mental health issues among German law students, with anxiety and depression being particularly prevalent. Our analysis also indicates that physical activity is significantly correlated with lower levels of depression, stress, and anxiety. This finding is consistent with previous studies, such as those by *Skead & Rogers (2016)* and *Skead, Rogers & Johnson (2020)*, which

demonstrate the protective effects of physical activity on mental health. Regular exercise appears to mitigate some of the psychological burdens experienced by law students, reinforcing the broader literature that supports physical activity as a crucial component of mental well-being.

The role of social relations and body image perception emerged as significant factors in our study. We found that positive social interactions are associated with lower levels of stress and anxiety, consistent with the research of *Baba & Bunji (2023)*, who emphasized the importance of social support. Similarly, poor body image perception was linked to higher levels of anxiety, aligning with findings from *Skead, Rogers & Doraisamy (2018)*. These results underline the importance of addressing both social and self-perception factors in mental health interventions.

Mood states and affective valence also showed significant correlations with mental health indicators in our study. These findings resonate with *Ayres et al. (2017)*, who highlighted the impact of emotional regulation on managing anxiety among law students. The correlation between mood states and mental health outcomes suggests that improving emotional well-being could play a critical role in alleviating mental health issues. This study confirms existing research on the significant mental health challenges faced by law students, emphasizing the critical roles of physical activity, social relations, and self-perception. Our findings uniquely highlight how these factors are interrelated and collectively influence mental health outcomes. The strong correlations observed suggest that interventions enhancing physical activity, social support, and self-perception could effectively improve mental health among law students. To address these challenges comprehensively, a holistic approach is essential. Future research should explore these correlations further in varied contexts and larger, more diverse populations to deepen our understanding and validate these findings.

## Prediction of mental health levels

Our regression analyses revealed that anxiety and stress are strong predictors of depression among law students. This finding is in line with *Mukta et al. (2022)*, who demonstrated that negative emotional states significantly impact overall well-being and academic performance. The interdependence of these mental health indicators in our study suggests that effective interventions should address anxiety and stress as primary targets to reduce depression levels. Physical activity was identified as a significant predictor of lower levels of depression, stress, and anxiety. This supports the findings of *Skead & Rogers (2016)* and *Şahin, Çekin & Yazıcılar Özçelik (2018)*, which highlight the mental health benefits of regular exercise. Our study extends these findings by confirming that physical activity is a crucial element of mental health strategies, especially in high-pressure environments like law school.

The negative association between social relations and mental health problems found in our study underscores the importance of fostering strong social connections. This aligns with *Baba & Bunji (2023)*, who noted that social support is crucial for managing stress and anxiety. Our results suggest that interventions designed to enhance social ties could be

effective in improving mental health among law students. Additionally, body image perception and mood states were significant predictors of anxiety. This finding is consistent with *Skead, Rogers & Doraisamy (2018)*, who found that body image concerns and mood dysregulation contribute to psychological distress. Our results indicate that addressing body image and mood management could be effective strategies for reducing anxiety levels, particularly in competitive academic environments.

Moreover, the predictive relationships identified in our study highlight the intertwined nature of stress, anxiety, and depression. This interconnection is well-documented in the literature; for example, *Hjorth et al. (2016)* found that mental health issues are strongly correlated with academic challenges and higher dropout rates. The overlap between these mental health indicators suggests that interventions should not only target individual symptoms but also address the underlying factors that contribute to overall psychological distress. Overall, the predictive relationships identified in our study highlight the interconnected nature of mental health indicators. The overlap between anxiety, stress, and depression suggests that a comprehensive approach addressing multiple factors is necessary for effective mental health interventions. The findings support the need for holistic strategies that encompass physical activity, social support, body image improvement, and emotional regulation.

Our study supports and expands upon existing research by identifying anxiety, stress, physical activity, social relations, and body image as key predictors of mental health outcomes among law students. The strong predictive relationships observed highlight the need for a holistic approach to mental health care that addresses various aspects of students' lives to foster overall well-being. This underscores the necessity for comprehensive strategies that target these areas simultaneously. To build on these insights, future research should explore these predictors in diverse contexts and larger populations, enhancing our understanding and guiding the development of tailored interventions for law students and other high-stress groups.

## LIMITATIONS

Despite its valuable contributions, this study has several limitations that must be addressed. The sample characteristics represent a significant constraint, as the research focused exclusively on first-year law students, which may limit the generalizability of the findings to other student populations or demographic groups. Future studies should incorporate more diverse samples to improve the external validity of the results. Additionally, the cross-sectional design of the study prevents the establishment of causal relationships between variables. Longitudinal research would be beneficial to understand the temporal dynamics and potential causal pathways between mental health indicators and quality of life measures.

The reliance on self-report measures, such as questionnaires assessing mental health and quality of life, introduces potential response biases and social desirability effects. Future research could enhance the reliability and validity of findings by incorporating objective measures or multi-method approaches. Furthermore, the study did not account for potential confounders such as socio-economic status, prior mental health history, or

specific academic stressors, which could influence the relationships between mental health indicators and quality of life measures. Considering these factors in future research would provide a more accurate estimation of the associations being investigated. Finally, the study's setting within a single institution may limit the generalizability of the findings to other educational contexts. Replicating the study in multiple institutions or across different cultural contexts would strengthen the robustness and external validity of the results. Recognizing these limitations is crucial for contextualizing the findings and directing future research efforts to further our understanding of the intricate interplay between mental health and quality of life among college students.

## CONCLUSIONS

This study has successfully explored the potential to predict mental health outcomes among first-year law students using key variables such as mood states, physical activity, body image perception, and social relations quality. Despite the limitations of a small sample size, our findings indicate that these factors are significant predictors of depression, stress, and anxiety. These findings underscore the complex interplay between mental health indicators and various dimensions of quality of life, emphasizing the necessity for a comprehensive approach to mental health care. By identifying these predictors, we have gained a clearer understanding of the factors that impact mental health in this specific population. The insights gained highlight the value of interventions aimed at improving mood, increasing physical activity, enhancing body image, and strengthening social connections. These targeted strategies could effectively address mental health issues and promote well-being among law students. Future research should further investigate these relationships and develop tailored interventions to better support students' mental health. This study contributes to understanding the complex interplay between mental health and quality of life, offering a foundation for both practical interventions and future research.

### Funding
The authors received no funding for this work.

### Competing Interests
The authors declare that they have no competing interests.

### Author Contributions
- Raul-Ioan Muntean conceived and designed the experiments, performed the experiments, authored or reviewed drafts of the article, writing—original draft preparation, and approved the final draft.
- Valentina Stefanica conceived and designed the experiments, analyzed the data, authored or reviewed drafts of the article, writing—original draft preparation, writing—review and editing, and approved the final draft.

- Daniel Rosu conceived and designed the experiments, analyzed the data, authored or reviewed drafts of the article, writing—original draft preparation, and approved the final draft.
- Alexandru Boncu performed the experiments, prepared figures and/or tables, project administration, and approved the final draft.
- Iulian Stoian performed the experiments, prepared figures and/or tables, project administration, and approved the final draft.
- Mihaela Oravitan conceived and designed the experiments, analyzed the data, authored or reviewed drafts of the article, project administration, and approved the final draft.

### Human Ethics

The following information was supplied relating to ethical approvals (*i.e.*, approving body and any reference numbers):

The study was conducted in accordance with the Declaration of Helsinki, and approved by the Institutional Review Board (or Ethics Committee) of Doctoral School of Physical Education and Sport Science of West University of Timisoara (08/21.03.2023). Prior to registration, all participants were provided with comprehensive information regarding the study's objectives, methodologies, potential hazards, and advantages.

### Data Availability

The raw measurements are available in the Supplemental Files.

### Supplemental Information

Supplemental information for this article can be found online at http://dx.doi.org/10.7717/peerj.18245#supplemental-information.

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
