# Peer review of "Examining the interplay between mental health indicators and quality of life measures among first-year law students: a cross-sectional study"

_PeerJ, doi:10.7717/peerj.18245_

## Round 0.1 · original submission · Major Revisions

Please revise your manuscript and on resubmission provide a point-by-point response to how things have been edited. In any cases where you have not made changes based on a reviewer comment, please provide justification for that decision.

INTRODUCTION AND DISCUSSION:

R1 and R2 have highlighted that in your introduction you have not discussed relevant prior literature investigating law student mental health. It is important to situate your research within the pre-existing literature on the topic. I recommend signficantly shortening the introductory material on the broad nature of relationships between certain factors and wellbeing to make room for much more coverage of the relevant prior literature. I recommend making a sub-heading titled something like 'Prior studies examining mental health of law students' and under that sub-heading discuss the literature on this topic. By doing this, you might be able to determine how your research contributes to the literature on this topic, and make an explicit statement about that under the 'research objectives' sub-section.

Read the introduction of this paper you will see many relevant papers cited that you can also look up, and discuss: Skead, N. K., Rogers, S. L., & Johnson, W. R. (2020). The role of place, people and perception in law student well-being. International Journal of Law and Psychiatry, 73, 101631.

All reviewers have highlighted that you have not discussed your research findings within the context of relevant background literature. By covering relevant literature in your introduction section as mentioned above, you can then refer back to that literature in your discussion section to address this criticism from reviewers.

Much of the material included in your discussion is not entirely relevant. Please edit to discuss your findings in the context of relevant prior background literature instead. When I say 'relevant' I am referring specifically to other prior studies that have investigated law student mental health.

METHOD:

The reviewers have highlighted that you have not provided adequate description of the questionnaires used in your research. Please provide that detail in your revision. I recommend looking up other survey research papers to see how this is typically done.

RESULTS:

I note that p = .000 is stated many times throughout your results. This should be changed to p < .05 (or alternatively p < .001). Please note that there are some statistical software programs that report p values as zero due to issues with presentation of the very small numbers. In reality, what gets reported as zero is not a zero value number, it is instead a very small number like 0.0000000000003456 or something similar.

Your regression analysis does not appear to be reported or interpreted as per standard practice for reporting such analysis. I recommend looking at other research papers that report on multiple regression analysis (and look up relevant chapters in statistics textbooks) to gain some insights into how this is done and modify accordingly. When you look into the background literature on papers on law student mental health you should come across a few of those studies that have reported a regression analysis. Note that in multiple regression analysis like what you have done, the standardised regression coefficients are more informative than unstandardised coefficients. You should report on both the unstandardised and standardised coeefficients. If you do not feel comfortable with reporting and interpreting regression analysis, you might wish to consider leaving that out of your paper on your revision, and stick to only reporting and discussing the correlations.

·

Basic reporting

no comment

Experimental design

no comment

Validity of the findings

no comment

Additional comments

Clarify Research Objectives: The research objectives need to be clearly defined and justified, particularly the second and third objectives. Explain how predictive analysis of depression will be approached given the small sample size and specify the types of interventions considered for the third objective.

Improve Consistency and Terminology: Ensure consistent use of terminology throughout the paper. For example, the terms "college students" and "university students" should not be used interchangeably unless there is a clear distinction being made. Consistency will help avoid confusion and enhance the clarity of the study.

Expand on Socio-Demographic Data: Include comprehensive socio-demographic information about the respondents in the results section. This data provides valuable context for understanding the key findings and enhances the overall robustness of the study.

·

Basic reporting

The article fails to cite relevant statements in the introduction and discusses its results without comparing them with existing literature.
The study also made use of old literature where current studies could suffice

Experimental design

No comment

Validity of the findings

The results are valid but should be compared with existing literature, comments have been placed within the document for revision

Additional comments

The article should be edited for grammatical correctness, there is also a need to add relevant sections as highlighted to reduce ambiguity.

·

Basic reporting

Review
Examining the interplay between mental health indicators and quality of life measures among first-year law students: a quantitative study
1. language is clear
2. introduction and background is good
3. literature is well referenced

Experimental design

1. LINE = 137-138: participants selected on April 17 and 24 from university of physical education and sports classes there is no mention how many were selected.
2. LINE= 160-161: 75 subjects selected at 1 Decembrie 1918 university of Alba Lulia, are they the same subjects or different subjects from different university, plz explain if they were different or same subjects, if they were different subjects then why and were they matched and how
3. LINE=165-166: it confusing for international reader as if there are three universities while third university of Timisura granting ethical approval, please clarify the confusion.
4. LINE=171: please elaborate DASS-21 on how many questions in the questionnaire, how much time consuming, how many questions for depression, anxiety and stress are specific in the questionnaire
5. LINE=179: please elaborate how EIFI is scored and how its score is interpreted.
6. LINE=183: please elaborate MBPAQ, how many question it contains, how it is scored and how its score is interpreted.
7. LINE=187-188: please elaborate WHOQoL-BRIEF, how many questions for each entity, how it is scored and how scores are interpreted.
8. LINE=191: please elaborate CBRS, how many questions it contains, how it is scored and how its results are interpreted.
9. LINE=210-211: please clarify which variable is independent and which is dependent.

Validity of the findings

ok

Additional comments

DISCUSSION:
a. Descriptive statistic and Correlation Analysis: under both of these headings results are portrayed which have already been displayed under the heading of RESULTS. There is no discussion or comparison indicating concordance or discordance of these results with the contemporary research work.
b. Prediction of Mental health Levels: please elaborate how the results from study of Mukta and study of Baba and Bunji correlate with the results of this study, instead of highlighting results of their study as a separate item please incorporate their result to emphasize and highlight results of this study. It needs to look like ingredient of your dish rather than a separate dish altogether.

---

## Round 0.2 · accepted · Accept

When going through the final tasks prior to production of the article, I recommend you consider editing your abstract results to list either correlation values or standardised beta values instead of mean values. This is because in that part of the abstract you are referring to correlational type results instead of mean camparison type results.

·

Basic reporting

no comment

Experimental design

no comment

Validity of the findings

no comment

Additional comments

Dear Author’s,

Thank you for your prompt and thoughtful revisions to the manuscript. I appreciate the effort you’ve put into addressing the suggestions and making improvements throughout the text. The clarity in the abstract, the inclusion of socio-demographic data, and the consistency in terminology have greatly enhanced the overall readability and coherence of your paper. Additionally, the adjustments to the research objectives, the explanation of emotional intelligence, and the expansion of the research tools section have added significant value to your methodology and results.

I believe the manuscript is now in a much stronger position, and I’m confident it will make an important contribution to the field.

Thank you once again for your hard work and dedication to this research. I look forward to seeing your future contributions.

·

Basic reporting

no comment

Experimental design

line 163: why giving ref of indu and vidhukumar 2019 it is your original study, even if there is need of giving ref over here then incorporate it in continous sentence like"this......studyfocusing on ......is simmilar to the study done by.........
line 214: how the sample size was calculated and based on which statistics
MATERIAL AND METHODS: under this heading there no need to show results please incorporate results under the heading of RESULT. material and methods is only for the detailedelaboration of materials and methods utized to conduct the study.
please mention the cut scores along with how interpreted.
results are scattered uner the heading of material and methods and with the tables but not under the heading of results

Validity of the findings

no comments

Additional comments

typing and commutation needs attention